# An examination of the association between early initiation of substance use and interrelated multilevel risk and protective factors among adolescents

Carlos Andres Trujillo[1☯], Diana Obando[2☯], Angela Trujillo[2☯] *

1 School of Management, Universidad de los Andes, Bogota, Colombia, 2 Department of Psychology, Universidad de La Sabana, Chía, Cundinamarca, Colombia

☯ These authors contributed equally to this work.
* angela.trujillo@unisabana.edu.co

**Data Availability Statement:** All relevant data are within the paper and its Supporting Information files.

## Abstract

One of the major goals of drug use prevention programs is to delay the age of onset of substance use. What is called early initiation, usually occurring in adolescents under the age of 15, is a salient predictor of Substance Use Disorders later in adulthood. The causes of early initiation are complex and multifaceted and this has led to the identification of a rich set of risk and protective factors that influence age of onset. Nonetheless, there is little knowledge about the interdependence of these factors in their impact on early initiation. This paper addresses this question by applying Multiple Correspondence Analysis to data on family, community and social risk and protective factors from over 1200 adolescents. We find that community and to a lesser extent social factors are the most clearly associated to early initiation and we compare our results to those obtained from linear regression analyses of the same data that do not incorporate interdependence and find opposite results. We discuss the differences between linear regressions and MCA to evaluate the interplay of risk and protective factors and the implications of our findings for health policy and the design of prevention interventions aimed at delaying age of onset.

## Introduction

Early initiation of substance use during adolescence is a salient predictor of Substance Use Disorders later in life [1] [2] [3] [4]. Thus, delaying the age of initiation is one of the major goals of prevention programs targeting the young. There are risk factors (RF) and protective factors (PF) that influence age of initiation [3] [5] [6] but there is no clear blueprint on how to prioritize actions targeted at diminishing RF and enhancing PF. One salient issue, empirically understudied in the literature, is the complex interdependence between RF and PF as they influence target behavior such as age of initiation. It is acknowledged that the interplay between RF and PF is highly relevant to explain specific behavioral patterns [7]. Jones, Hill, Epstein, Lee, Hawkins & Catalano [8] pointed out the Social Development Model and

**Funding:** The author(s) received no specific funding for this work.

**Competing interests:** The authors have declared that no competing interests exist.

developmental cascades framework as valid theoretical foundations based on which to understand the interrelation of RF and PF holistically. Similarly, ecodevelopental theory promotes the integration of RF and PF to uncover multiple determinants of substance use initiation among young adolescents [9]. However, the common approach reported in the literature to analyze the influence of RF and PF on relevant behaviors consists of conducting different types of multivariate regression analyses, and using the size of regression coefficients as an indicator of the power of RF and PF to influence the target variable (e.g., age of initiation). Usually, the size of regression coefficients captures the effect of each independent factor when other factors are held constant at their average. That is, they show the "isolated" effect of each factor. Some statistical interactions are frequently reported but to our knowledge, there is no account of empirical work looking at the combined interactive effects of multiple RF and PF. In addition, most regression models are based on the assumption of linear relationships between independent and dependent variables, which is arguably an oversimplification of the complex links among RF and PF. Consequently, given the interdependent nature of RF and PF and the potential nonlinearity of their linkages, multivariate regressions may be offering an inexact blueprint of how to prioritize RF and PF in interventions targeted at reducing age of initiation. In this paper we tackle this issue by means of a different analytical approach. We examine a multivariate interdependence technique (Multiple Correspondence Analysis, MCA) as an alternative way to answer how to prioritize RF and PF in order to influence age of initiation. MCA is able to capture multiple associations among interdependent factors without making assumptions about functional forms (linear or otherwise) [10]. We include RFs and PFs at the social, community, and family level. We compare our results with that of multiple regression models for the same data and discuss the differences and derived implications. In short, we find that the two statistical approaches indicate a different order of associations between RF, PF and early initiation.

## Early initiation of drug use as a predictor of later adolescent and adult drug use disorders

Using drugs and alcohol at an early age has been associated with multiple problems later in life such as negative health, social, and behavioral outcomes [11] [12]. Kessler et al. [13] have suggested that age of onset should be a major focus of study because it explains the risk of a disorder in the future. There are an extensive number of studies that show that initiation of substance consumption early in life influences the levels of use and abuse in later life as well as alcohol abuse disorders. For example, DeWitt et al. [5] reported that the likelihood of developing lifetime alcohol dependence increases with each year of alcohol use onset age. Liang and Chikritzhs [14] found associations between early onset age and later heavy drinking; the authors suggest that this happens because the earlier the age of onset, the longer the time at risk for alcohol consumption to escalate until it becomes a problem. In addition, early substance initiation has been identified as an RF for injection drug use [6], binge drinking [15], hazardous levels of alcohol consumption and drug use [16]. Epidemiological studies found evidence of higher alcohol dependence rates for individuals that initiated alcohol use by age 14, than those who started at 20 or older, as well as illicit drug dependence or abuse for those who first tried marijuana at age 14 or younger [3]. Early-onset cocaine users have also shown neuropsychological alterations and higher polydrug use, than those with a late-onset [17]. Early cannabis use has been associated with other drug use and substance use disorders. The prevalence of drug abuse symptoms in adulthood decreased with later age of cannabis use onset. Rioux, Castellanos-Ryan, Parent, Vitaro, Tremblay and Séguin [18] found a difference

of 30% in the presence of these symptoms between those who started at 13 years or earlier and those who started using cannabis at 17 years when they were 28.

## Risk and protective factors associated with age of initiation

Despite the relevance of early initiation, scarce evidence is available about its causes. According to Kaplow, Curran and Dodge [19], it is relevant to identify whether the early onset of drug use and the eventual substance abuse involve the same associated factors. A small number of studies have explored specific RF and PF associated with early engagement with drugs. As such, Malmberg, Overbeek, Monshouwer, Lammers, Vollebergh and Engels [20] found associations between early adolescence symptoms of anxiety, hopelessness, sensation seeking and impulsivity, and lifetime prevalence and age of onset for different drugs. Moreover, Kaplow et al. [19] identified that during childhood, personal factors such as overactivity, thought problems and problem solving skills, and parental substance abuse contributed to early initiation of drug use. In this line, a recent study from Maggs, Staff, Patrick and Wray-Lake [21] pointed out that personal and parental factors, such as parent lack of control of children's behavior and parent substance use are specific predictors of early initiation of alcohol consumption. A study conducted with Indigenous youth from the US and Canada revealed that positive representations of the prototypical adolescent drinker and having peers who drink increases the risk for the onset of alcohol use [22]. Aggressive behavior, gender, and father's educational level have also been identified as predictors of early alcohol use [23]. Initiating sipping or tasting alcohol was predicted by perceived parental approval and by current parental drinking status, influencing attitudes toward this behavior [24]. A family history of alcohol problems has also been associated with initiating drinking [25] [16] and its effect is stronger before the age of 15.

Other studies have explored the contributions of broader domains on early substance initiation. For example, Burlew, Johnson, Flowers, Peteet, Griffith-Henry and Buchanan [26] identified among African American youth that living in a community characterized by high levels of substance use, violence and poverty increases the risk for the early onset of drug use. Also, promotional alcohol items encourage alcohol initiation [11]. Enstad, Pedersen, Nilsen and von Soest [27] assessed personal, social, economic and family dimensions to identify specific predictors for early onset of intoxication compared to early onset of drinking behavior. This study involving Norwegian adolescents showed that temperament, norm-breaking behavior, socioeconomic features, and family factors predicted early onset of intoxication, while low levels of shyness and high friend deviancy were associated with early onset of drinking.

PF associated with substance use age of onset has also been explored in previous studies. However, less evidence is found in this regard. In a sample of Mexican youngsters, Atherton, Conger, Ferrer and Robins [28] found that close families with strong values show a decreased risk of early engagement with drugs. Ryan, Jorm and Lubman [29] argued that delayed alcohol initiation was predicted by parental drinking modeling, by limiting the access of alcohol to children, by quality parent-child relationships, and by parental involvement and communication. Parental monitoring and warmth are indirect antecedents of drug onset, as they predict adolescents' social perception of drug use [30]. Family attachment also indirectly increases the age of drug use onset as it lowers negative symptoms and sensation seeking in adolescents [31]. In addition, Bacio et al [9] argued that late initiation of drug use is an outcome of positive school climate, which lowers perceptions of norms of use between peers, reducing their likelihood of beginning substance use in early adolescence.

There are also categories of PF that influence a wider range of drug related behaviors (initiation, frequency of use, etc.), including family, school and community, which are the focus of this work. In the family category, substance is reduced by family connectedness [32] [33],

parental supervision [34], clear rules [35], positive parenting style [36], and living in a two-parent family [37]. At school and community levels, some factors that have been shown to lower the probability of drug use among adolescents are: positive climate at school, connection to school or other adults in the neighborhood, feeling safe at school or in the community, policies and practices that support health, norms, and opportunities in the community for meaningful engagement [36] [37]. This knowledge, however, falls short in explaining the interdependence of factors. Moreover, interdependence may alter the way in which each individual factor influences drug related behavior, among which early initiation stands out as critical in the prevention of later SUDs. Hence, we argue that understanding the interplay between RF and PF at all levels of adolescents' socialization is essential to developing well-targeted prevention strategies to delay the age of onset. The objective of this study is to contribute to that goal, by identifying the RFs and PFs that are more closely associated to early initiation, accounting for the interplay of social, community, and family factors.

## Method

We analyzed the relationship between age of onset and RF and PF using cross-sectional data. We applied a survey to 1272 adolescents aged between 12 and 19. The mean age was 14.87 (SD = 1.31), and 56% were girls. The Research and Ethics Committee at Universidad de la Sabana approved the research procedures for data collection for this study in minute number 62, 2013. We used the Spanish version of the Communities That Care Youth Survey (CTC-YS) by Arthur, Hawkins, Pollard, Catalano and Baglioni [38]. Professionals trained in the application of the CTC-YS administered the electronic version of the survey at the schools. This electronic version was set up such that all questions had to be answered in order to allow participants to progress along the questionnaire. Hence, no missing data was generated. To guarantee confidentiality, we used codes for each participant. Electronic responses were recorded in an excel file and then exported for later statistical analyses.

The CTC—YS test consists of 135 questions. It measures the level of exposure to both RF and PF related to the consumption of alcohol, cigarettes, marijuana, inhalants, and other substances. It also measures consumption patterns and antisocial behaviors among adolescent students in grades 6 to 12. RF and PF are measured at different levels: family, social, community, and personal characteristics and beliefs. We focused on factors grouped by a) Family: Opportunities ($\alpha = 0.75$) and rewards ($\alpha = 0.62$) for prosocial involvement in the family, family attachment ($\alpha = 0.80$), family conflict ($\alpha = 0.69$), family history of antisocial behavior ($\alpha = 0.74$), favorable attitude toward substance ($\alpha = 0.44$) and poor family management ($\alpha = 0.81$); b) Social dynamics: school perceived opportunities ($\alpha = 0.58$) and rewards ($\alpha = 0.67$) for prosocial involvement, low commitment to school ($\alpha = 0.65$) and bad relations with peers ($\alpha = 0.60$); and c) Community: laws and norms favorable to drug use ($\alpha = 0.71$), community disorganization ($\alpha = 0.66$), low neighborhood attachment ($\alpha = 0.80$), perceived availability of drugs ($\alpha = 0.77$), opportunities ($\alpha = 0.66$) and rewards ($\alpha = 0.79$) for community prosocial involvement. In total, 17 RF and PF were assessed by asking adolescents in a 4-point scale (1 = definitely not true, 4 = definitely true) about their levels of exposure to both RF and PS related to substance use.

In the CTC-YS, early initiation is measured using the following six items: How old were you when you first. . . smoked a cigarette, even just a puff; had more than a sip or two of beer, wine or hard liquor (vodka, whiskey or gin); began drinking alcoholic beverages regularly, that is, at least once or twice a month; smoked marijuana; used inhalants (gasoline, glues, among others); the first time you got drunk. The survey was implemented by means of a collaboration effort with the municipalities of Cogua and Ubaté, near Bogota. The study objectives were

explained to schools in both municipalities. The schools that expressed interest in participating sent informed consents to the adolescents' parents or caregivers for them to be included in the sample. Adolescents whose parents signed the informed consent were asked to ascertain their participation by signing an assent consent.

As explained earlier, we sought to evaluate the association between early initiation in drugs and the 17 RF and PF from a perspective that 1. Makes no assumption about the mathematical functional form of the relationships and 2. Incorporates the interdependent nature of RF and PF in the analysis. In order to do so, we chose to use a Multiple Correspondence Analysis (MCA) as a statistical tool. This is an interdependence technique that establishes the association between categorical variables based on the mathematical (Chi-square) distance between categories and objects within these categories [10] [39] [40]. It is a compositional technique that provides a low dimensional representation of the multiple associations of all possible two-way cross-tabulations of a set of categorical variables. The geometrical distance between categories and category levels is a standardized measure of association based on the conditional probability of observations of a category *a*, given another category *b*.

Table 1 shows basic descriptive statistics for RF, PF, and early initiation. We first look at the distribution of each of the RF and PF constructs including the measure of early initiation. The composite measure of early initiation gives values from 0 to 7, where 0 is never, 1 = 17 years old and so on until 7 = 11 years old. Thus, the mean of 1.30 refers to an average age of between 16 and 17 years old. For the specific substances that constitute our measure, initiation of alcohol use was 3.70 (between 14 and 15 years old); of cigarettes, it was 1.14 (between 16 and 17 years old); and of marihuana, it was 0.32 (over 17 or never). As it pertains to the other RFs and PFs, the CTC-YS uses four-level ordinal-categorical items to capture RFs and PFs that are later averaged to obtain a more nuanced, metric measure of each RF and PF (a detailed list of items can be obtained from test authors). For the upcoming procedures and analyses, we use the Multiple Correspondence program embedded in SPSS v. 24.

The first step in the implementation of the MCA analysis is to discretize the RF and PF measures in three categorical levels, based on the mean and standard deviation of each RP and PF. This step gives a clear qualitative interpretation of the different categorical levels. For instance, our target variable, early initiation, is discretized in the following way: early initiation = between 11 and 14 years old (n = 288) (i.e., < 15); middle initiation = 15 and 16 years old (n = 612); and late initiation = over 17 years old or never (n = 372). Adequate cut off points to categorize early initiation are variable in the literature and dependent on the substance studied, In this study, considering that our measure includes various substances, the categorization is consistent with previously used theoretical cut off points for early initiation (see Donovan and Molina, 2011 [24]. (Supplementary material S1 contains the details of cutting points and interpretation of categories for all RF and PF). In all cases, we used a three-level ordinal categorization for a straightforward interpretation of their meaning and association. We conducted the analysis with all 17 RF and PF plus early initiation using variable principal normalization. We produced several partial biplots to facilitate interpretation, considering that in MCA there is no definition of dependent and independent variables. All two-way cross-tabulations are processed simultaneously.

## Results

To examine the relationship between RF and PF and early initiation, we focused the analysis on the relative position of the three early initiation categories vis-à-vis the 17 RF and PF. The two-dimension MCA can account for over 36% of the inertia, the first dimension accounts for

**Table 1. Descriptive statistics for all risk and protective factors.**

| | N | Cronbach's alpha | Min | Max | Mean | Std. Dev. |
|---|---|---|---|---|---|---|
| **Community** | | | | | | |
| Laws and norms favorable to drug use | 1272 | .71 | 1.00 | 3.86 | 2.21 | .50 |
| Community rewards for prosocial involvement | 1272 | .79 | 1.00 | 4.00 | 2.22 | .81 |
| Community opportunities for prosocial involvement | 1272 | .66 | 1.00 | 4.00 | 2.76 | .81 |
| School opportunities for prosocial involvement | 1272 | .58 | 1.00 | 4.00 | 2.80 | .49 |
| Community disorganization | 1272 | .66 | 1.00 | 3.67 | 1.94 | .54 |
| **Social** | | | | | | |
| Perceived availability of drugs | 1272 | .77 | 1.00 | 4.00 | 2.03 | .78 |
| School–negative relations with peers at school | 1272 | .60 | .00 | 4.00 | .86 | .60 |
| Low commitment to school | 1272 | .65 | 1.00 | 4.10 | 2.03 | .49 |
| Low neighborhood attachment | 1272 | .81 | 1.00 | 4.00 | 2.00 | .81 |
| School rewards for prosocial involvement | 1272 | .67 | 1.00 | 4.00 | 2.82 | .58 |
| **Family** | | | | | | |
| Early initiation of drug use | 1272 | .85 | .00 | 7.43 | 1.30 | 1.27 |
| Family attachment | 1272 | .80 | 1.00 | 4.00 | 2.89 | .65 |
| Family opportunities for prosocial involvement | 1272 | .75 | 1.00 | 4.00 | 3.11 | .72 |
| Family rewards for prosocial involvement | 1272 | .62 | 1.00 | 4.00 | 2.90 | .73 |
| Family history of antisocial behavior | 1272 | .74 | .63 | 3.75 | 1.58 | .62 |
| Family conflict | 1272 | .74 | 1.00 | 3.86 | 1.97 | .58 |
| Favorable parental attitudes toward drug use | 1272 | .44 | 1.00 | 4.00 | 1.38 | .53 |
| Poor family management | 1272 | .81 | 1.00 | 4.00 | 1.69 | .56 |

23% and the second for 13%, with each dimension displaying appropriate Cronbach's alphas (see Table 2).

## Data visualization

Given the high number of variables in the analysis, we will produce several partial biplots in order to facilitate the visualization of results. All figures come from one single MCA analysis and therefore this partial visualization does not affect the estimated distances between category levels and the accounted inertia is the same for all figures. In Fig 1 we show the full biplot of the RF and PF categories. Early initiation is highlighted as the bold circle. The first result of this visualization is that all RF and PF categories show an expected overall pattern of monotonic relation with early initiation. That is, high risk and low protection categories are grouped closer to early initiation, while low risk and high protection categories are grouped closer to late initiation. The pattern is the same for all RF and PF. The strength of association among categories is given by their geometric proximity.

**Table 2. Summary statistics of multiple correspondence analysis.**

| Dimension | Cronbach's Alpha | Variance Accounted For | | |
|---|---|---|---|---|
| | | Total Eigenvalue | Inertia | % of Variance |
| 1 | .809 | 4.244 | .236 | 23.578 |
| 2 | .606 | 2.338 | .130 | 12.989 |
| Total | | 6.582 | .366 | |
| Mean | .737 | 3.291 | .183 | 18.283 |

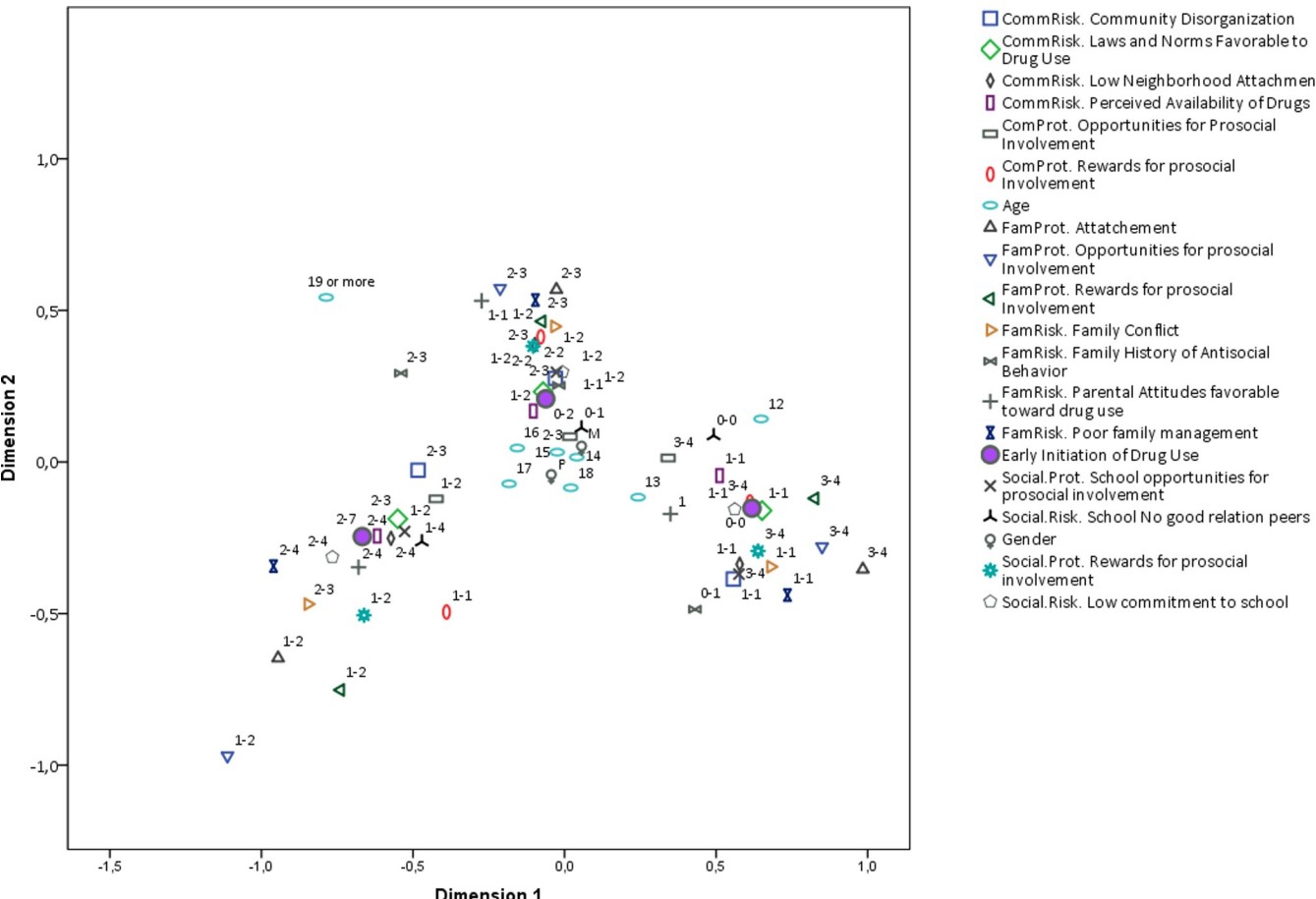

**Fig 1. Biplot of all risk and protective factor categories.** Early initiation highlighted *Interpretation note: All RF and PF were discretized in three ordinal categories. The labels in the plot region next to each category indicate the range of values included in such category. For instance, the label (1–2) next to a point in the plot means that such category point represents values from 0 to 1 the original scale of the corresponding RF or PF. Early initiation is the only variable that is plotted with a solid shape. Gender is represented by the letters F or M and Age is not discretized in the plot. It takes values from 11 to "19 or more".*

We will now present partial plots with fewer categories to improve interpretation. The first is a comparison of RF factors and early initiation (Fig 2) against PF and early initiation (Fig 3). There is a noticeable difference between the two plots. RF and early initiation category clusters are better defined than those of PF and early initiation, particularly for early initiation. This means that the pattern of association between RF and early initiation is more homogenous across categories than that of PF and early initiation. This result is consistent with the literature on RF and PF, where the effects of RF are reported to be stronger and more pervasive. However, the direction of the categories of the PF categories cluster indicates that a potential cause of such a difference is a nonlinear relationship between most PF and early initiation. This can be seen in the shape of the cluster around early initiation.

The general biplot can also be split by the dimension of RF and PF, namely family, social, and community. Figs 4, 5 and 6 contain the corresponding biplots. They show that the pattern of association is most homogeneous (and linear) for both RF and PF related to the social dynamic (Fig 4) in which the clusters around early, middle, and late early initiation are clearly separated from each other and their shape is rather rounded. The biplot for the family dimension is much less homogeneous (Fig 5), and once again the non-linearity is observed in the pattern of associations from middle to early initiation. Finally, the biplot for the community-

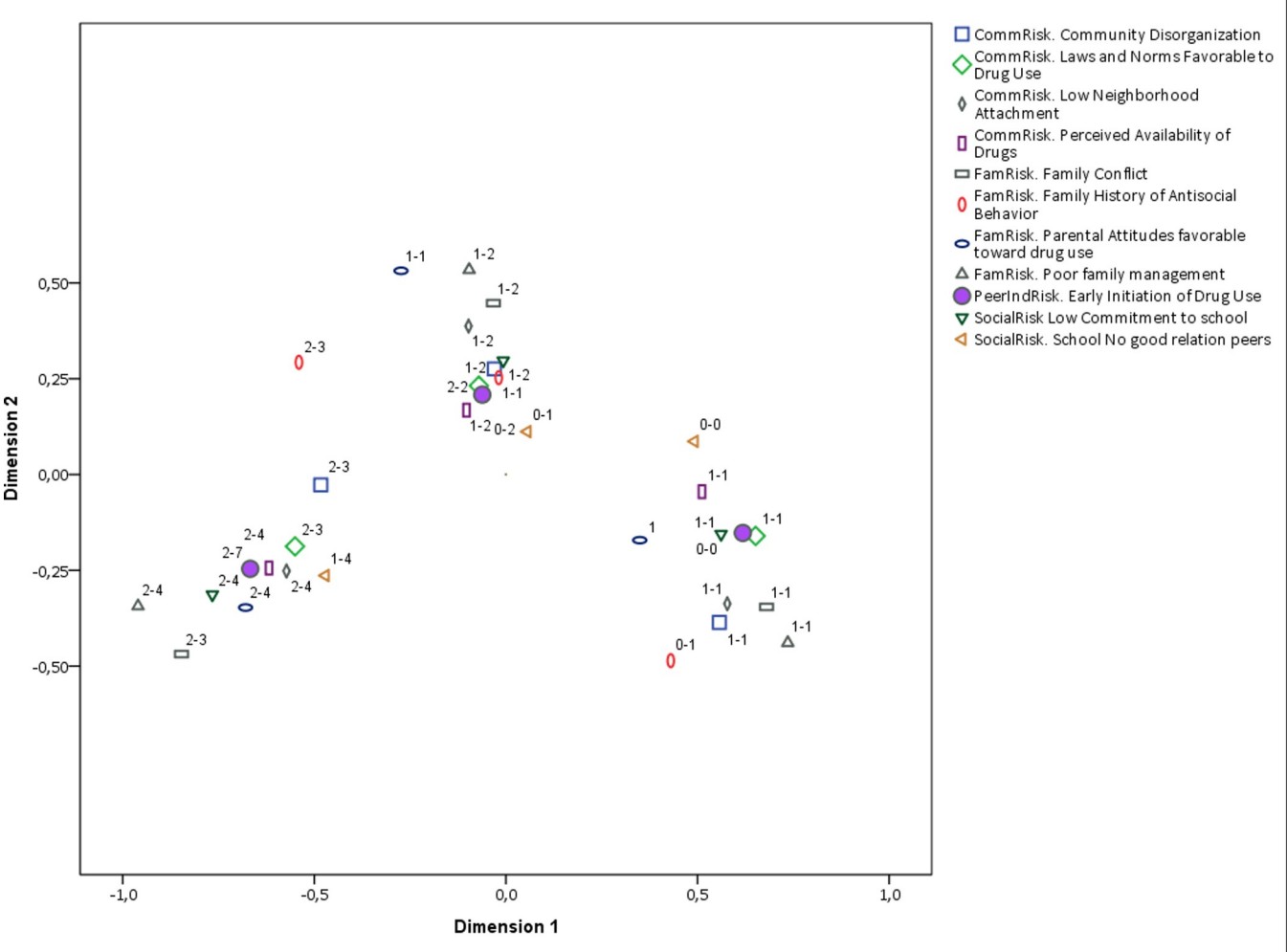

**Fig 2. Biplot of risk factor categories and early initiation.** Interpretation note: All RF and PF were discretized in three ordinal categories. The labels in the plot region next to each category indicate the range of values included in such category. For instance, the label (1–2) next to a point in the plot means that such category point represents values from 0 to 1 the original scale of the corresponding RF or PF.Early initiation is the only variable that is plotted with a solid shape.

related categories shows an even less linear relationship across all ordinal levels (Fig 6). The distances among RF, PF, and early initiation are differently distributed in each level. For some RF and PF, the level of early initiation they are most closely related to becomes visually unclear.

The next step is to look at the specific distances from every RF and PF to early initiation. As explained earlier, the closer the geometrical distance, the closer the association. This analysis also reveals the strength of the non-linearity between RF, PF and early initiation levels by calculating the absolute value of the normalized distances between each RF and PF category and the corresponding early initiation level, using the normalized coordinates in each dimension:

$$Total\ distance = \sum\nolimits_{i=1}^{3} |\ (F_{id1} - EI_{id1}) + (F_{id2} - EI_{id2})| \qquad\qquad \text{Eq 1}$$

where Fs are the different RF and PF, *d1* and *d2* are dimension 1 and 2 of the biplot, and i, where $i \in \{1, 2, 3\}$ is each of the three ordinal categorical levels. Table 3 contains the distances

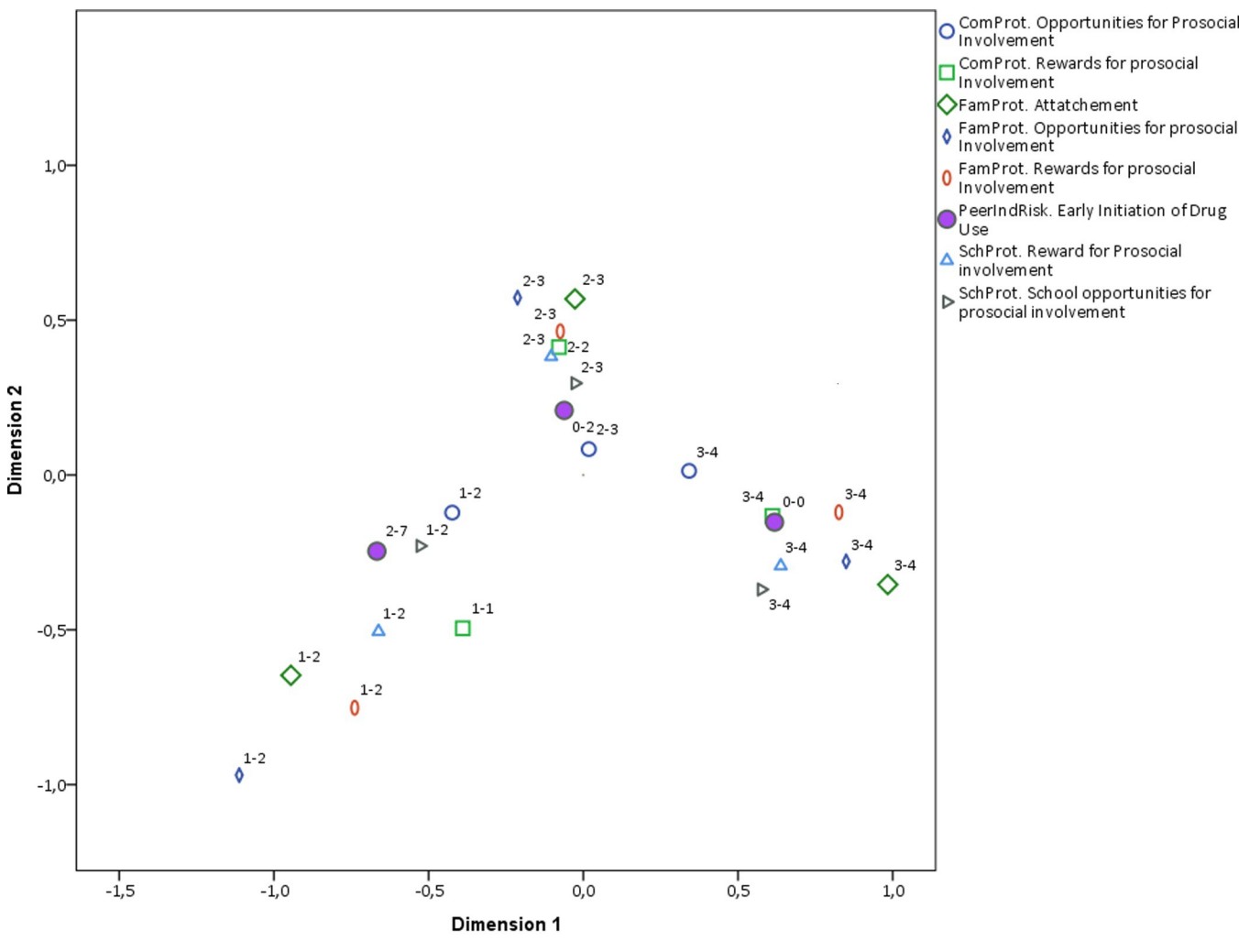

**Fig 3. Biplot of protective factor categories and early initiation.** Interpretation note: All RF and PF were discretized in three ordinal categories. The labels in the plot region next to each category indicate the range of values included in such category. For instance, the label (1–2) next to a point in the plot means that such category point represents values from 0 to 1 the original scale of the corresponding RF or PF. Early initiation is the only variable that is plotted with a solid shape.

for each RF and PF for each ordinal level and the total distance. If the three distances for each F are similar, this indicates linearity, if they vary it shows non-linearity. Table 3 is ordered from the most proximal to the most distal factors based on total distance. This means that RF and PF are ordered from the most closely associated to the least associated to early initiation. Considering that the biplot coordinates are calculated taking into account all two-way cross tabulations (and conditional probabilities) of every RF, and PF with each other, hence, the interdependence of factors is part of the results.

These results would indicate that in the population and area that this sample represents, interventions should focus mostly on social dynamics and community, which are the top five most closely associated factors with early initiation. The family dimension RFs and PFs are the least associated. In particular, the top five associations are to 1) perceived availability of drugs (RF), 2) laws and norms favorable to drug use (RF), 3) rewards for prosocial involvement in the community (PF), 4) negative relationships with peers at school (RF), and 5) low

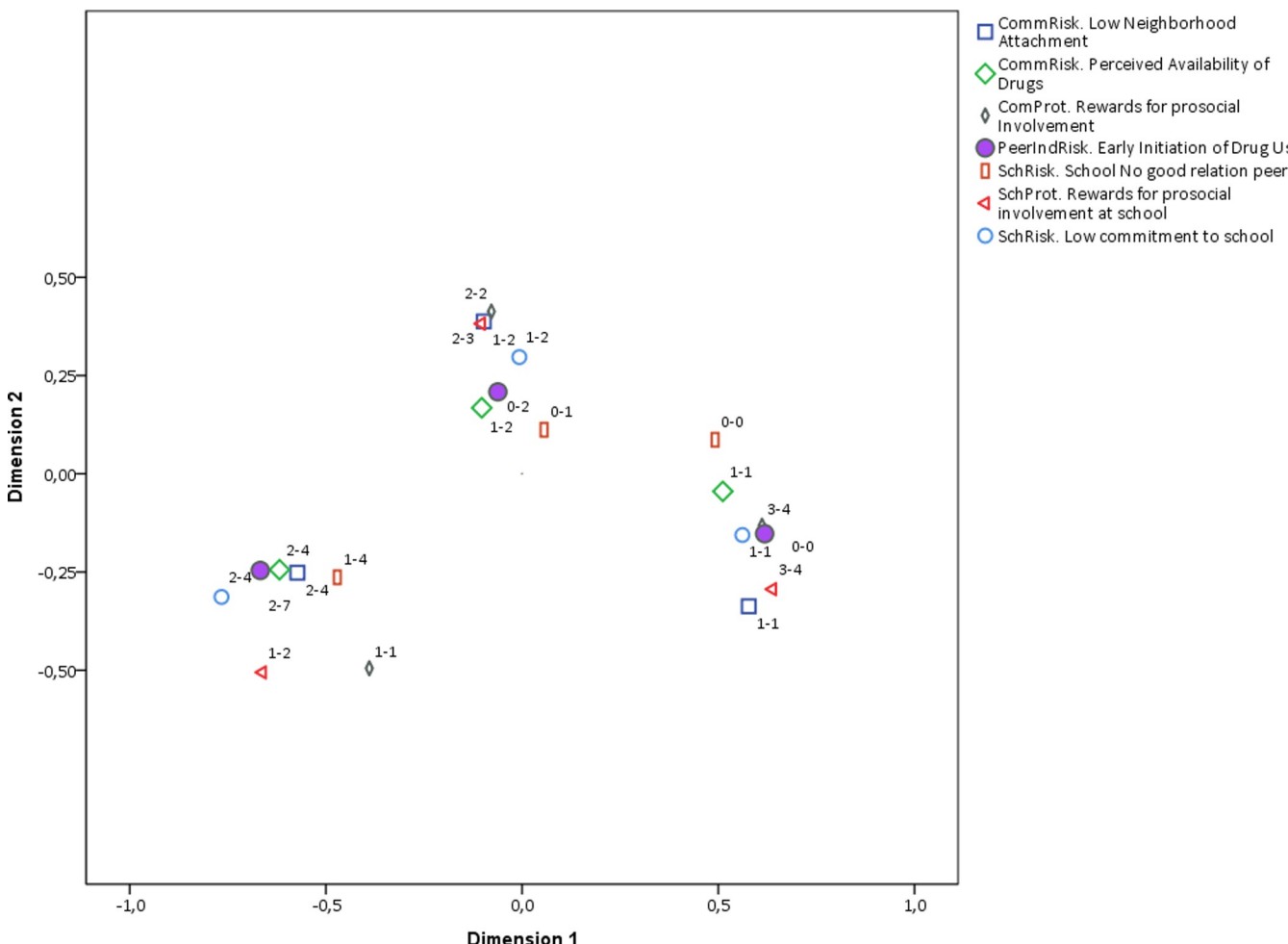

**Fig 4. Biplot of social dynamics categories and early initiation.** Interpretation note: All RF and PF were discretized in three ordinal categories. The labels in the plot region next to each category indicate the range of values included in such category. For instance, the label (1–2) next to a point in the plot means that such category point represents values from 0 to 1 the original scale of the corresponding RF or PF. Early initiation is the only variable that is plotted with a solid shape.

commitment to school (RF). The three least associated (i.e., more geometrically distal) are 1) family attachment (PF), 2) family history of antisocial behavior (RF), and 3) opportunities for prosocial involvement in the family (PF). It must be said that these conclusions are the outcome of the relative total of geometric distances. No effect sizes can be derived from this information.

To assess these results and the ensuing recommendations for prioritizing interventions, we also conduct a traditional analysis using multiple linear regression with early initiation as the dependent variable and all the RFs and PFs plus age and gender as independent variables. We estimated the following regression model:

$$EI = \beta_0$$

$$+ \sum_{i=1}^{18} \beta_i F_i^{P,R} + \beta_{gender}(age) + \beta_{gender}(gender) + \varepsilon \quad where \quad F_i^{P,R} \in \{18\,RF, PF\} \text{ Eq 2}$$

The regression model was able to account for 26% of variance (Adjusted R-squared);

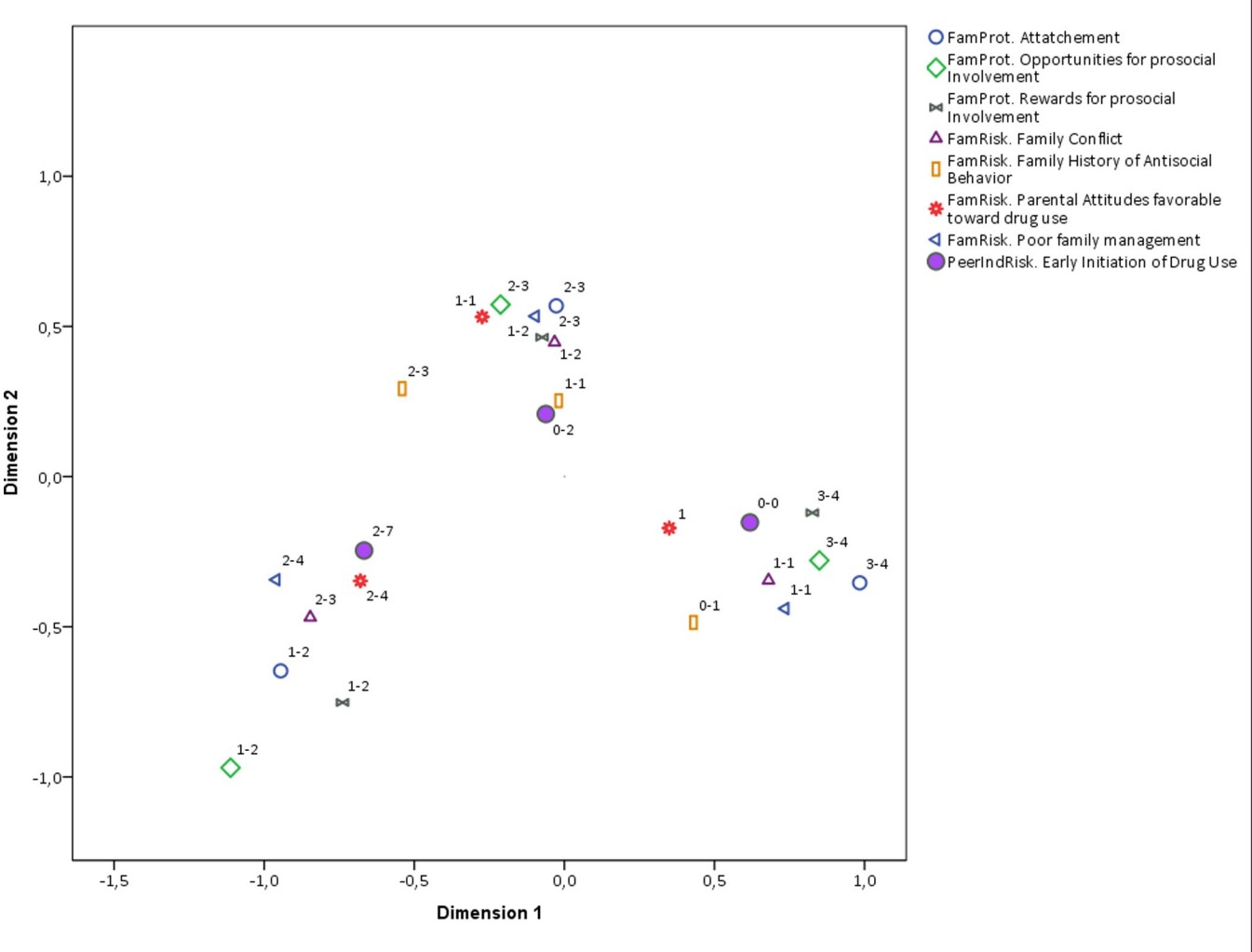

**Fig 5. Biplot of family categories and early initiation.** Interpretation note: All RF and PF were discretized in three ordinal categories. The labels in the plot region next to each category indicate the range of values included in such category. For instance, the label (1–2) next to a point in the plot means that such category point represents values from 0 to 1 the original scale of the corresponding RF or PF. Early initiation is the only variable that is plotted with a solid shape.

(F = 24.69, p = .00). In Table 4, we show the regression coefficients ordered by standardized betas to allow them to be compared. Effect sizes can be estimated using regression analysis, in this case using the standardized betas. They show that the five biggest effects are found for 1) parental positive attitudes towards drug use (RF), 2) perceived availability of drugs (RF), 3) family conflict (RF), 4) family history of antisocial behavior (RF), and 5) poor family management (RF). The four smallest effects are found for 1) opportunities for prosocial involvement in the community (PF) (31 times smaller than parental positive attitudes to drug use), 2) negative relationships with peers at school (RF), 3) laws and norms favorable to drug use (RF), and 4) opportunities for prosocial involvement at school (PF). In order to check the regression results for robustness, we also conducted an ordered logistic regression using the discretized variable of age of onset, as used in the MCA, and the same independent variables (RF and PF)

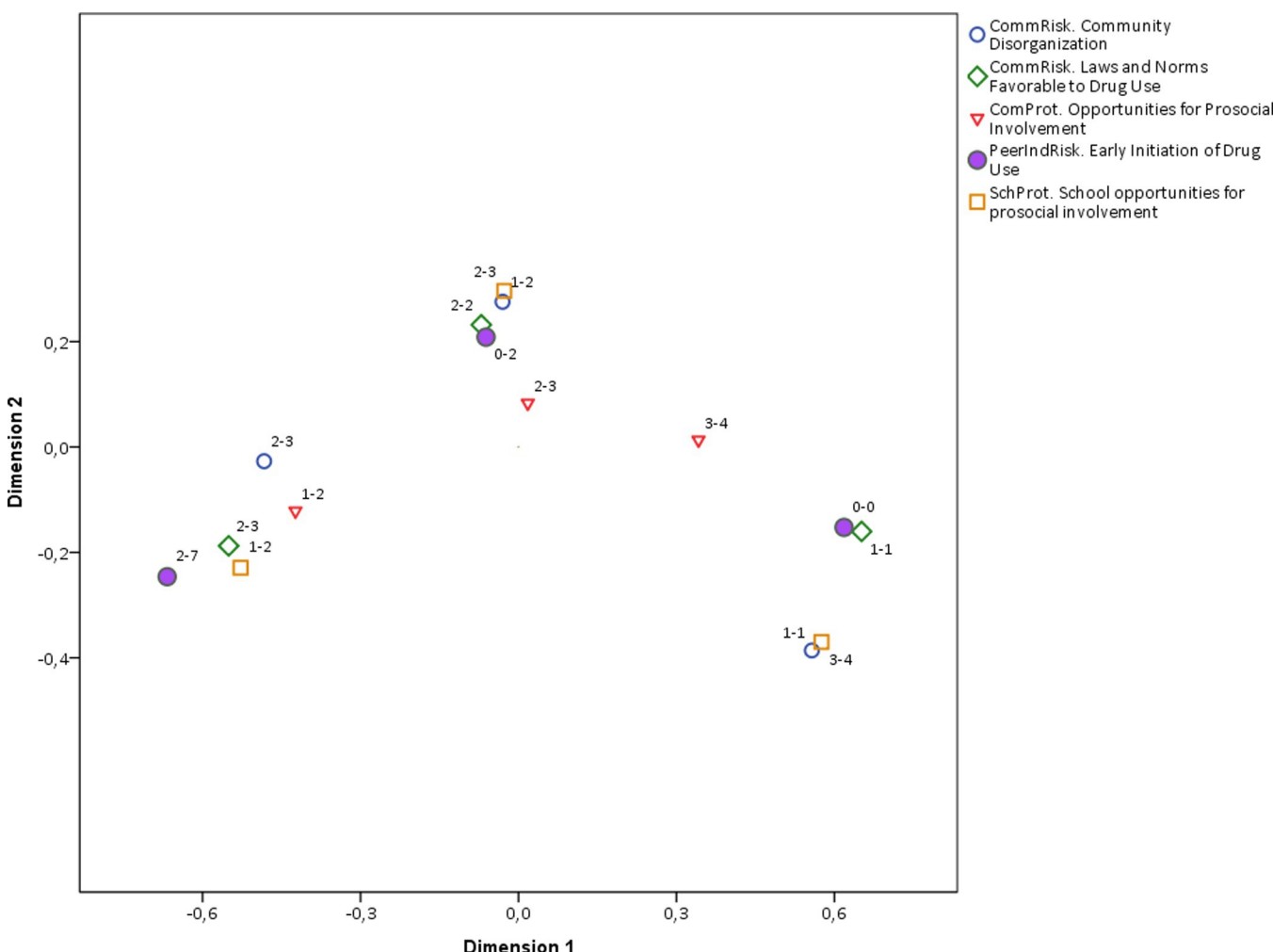

**Fig 6. Biplot of community categories and early initiation.** Interpretation note: All RF and PF were discretized in three ordinal categories. The labels in the plot region next to each category indicate the range of values included in such category. For instance, the label (1–2) next to a point in the plot means that such category point represents values from 0 to 1 the original scale of the corresponding RF or PF. Early initiation is the only variable that is plotted with a solid shape.

and covariates. Results were highly consistent with the linear regression. The ordered logistic regression details can be found in supplementary material S2.

## Discussion

The National Institute of Alcohol Abuse and Alcoholism (NIAAA) [41] affirms that prevention programs and public health policies should specifically target children under 15 and underage drinking, in general, in order to attempt to delay the onset of drinking alcohol as long as possible. The effects of early initiation on later life SUDs are very well documented. Moreover, debates about early initiation have important implications for the allocation of screening and prevention resources. Selected interventions during adolescence may benefit multiple domains including academic, peer relations, and delinquency [21]. For instance, knowledge about factors that delay early initiation has been essential in shaping alcohol prevention programs [27] [42]. This paper investigates the association between early initiation of substance use and RF and PF divided into family, social, and community categories. Our

**Table 3. Distances from RF and PF categories to early initiation, ordered from closest to furthest.**

| | | Distances from equivalent categories | | | |
|---|---|---|---|---|---|
| | | Early onset (< 15) | Middle onset (15–17) | Late onset (> 17 or never) | Total distance |
| Perceived availability of drugs | Social | 0.051 | 0.082 | 0.001 | 0.134 |
| Laws and norms favorable to drug use | Community | 0.175 | 0.014 | 0.026 | 0.215 |
| Rewards for prosocial involvement | Community | 0.029 | 0.188 | 0.014 | 0.230 |
| School–negative relations with peers | Social | 0.179 | 0.021 | 0.113 | 0.313 |
| Low commitment to school | Social | 0.166 | 0.143 | 0.060 | 0.370 |
| Low neighborhood attachment | Social | 0.089 | 0.143 | 0.225 | 0.457 |
| Reward for prosocial involvement | Social | 0.254 | 0.131 | 0.121 | 0.506 |
| Favorable parental attitudes toward drug use | Family | 0.113 | 0.111 | 0.288 | 0.512 |
| Opportunities for prosocial involvement | Community | 0.368 | 0.046 | 0.111 | 0.524 |
| School opportunities for prosocial involvement | Community | 0.156 | 0.123 | 0.260 | 0.539 |
| Community disorganization | Community | 0.403 | 0.099 | 0.295 | 0.796 |
| Family conflict | Family | 0.402 | 0.268 | 0.131 | 0.801 |
| Poor family management | Family | 0.390 | 0.291 | 0.169 | 0.851 |
| Rewards for prosocial involvement | Family | 0.578 | 0.243 | 0.240 | 1.060 |
| Attachment | Family | 0.679 | 0.395 | 0.165 | 1.238 |
| Family history of antisocial behavior | Family | 0.666 | 0.087 | 0.521 | 1.275 |
| Opportunities for prosocial involvement | Family | 1.168 | 0.213 | 0.105 | 1.486 |

findings therefore contribute to prior theory and research that calls for a multiple factor, multiple domain perspective to advance the understanding of social and environmental influences on youth well-being and adaptation [43] [44][45][46]. We used an approach that incorporates

**Table 4. OLS regression of early initiation on RFs and PFs (coefficients ordered by standardized betas).**

| | | Unstandardized Coefficients | | Standardized Coefficients | | |
|---|---|---|---|---|---|---|
| | Type | B | Std. Error | Beta | t | Sig. |
| Favorable parental attitudes toward drug use | Family | 0.444 | 0.062 | 0.189 | 7.116 | 0.000 |
| Perceived availability of drugs | Social | 0.302 | 0.045 | 0.187 | 6.722 | 0.000 |
| Family conflict | Family | 0.282 | 0.065 | 0.130 | 4.364 | 0.000 |
| Family history of antisocial behavior | Family | 0.235 | 0.055 | 0.116 | 4.285 | 0.000 |
| Poor family management | Family | 0.237 | 0.071 | 0.106 | 3.358 | 0.001 |
| Opportunities for prosocial involvement | Family | 0.176 | 0.070 | 0.101 | 2.524 | 0.012 |
| Low commitment to school | Social | 0.247 | 0.077 | 0.097 | 3.221 | 0.001 |
| Gender | | 0.212 | 0.065 | 0.083 | 3.249 | 0.001 |
| Community disorganization | Community | 0.116 | 0.064 | 0.050 | 1.818 | 0.069 |
| Reward for prosocial involvement | Social | 0.086 | 0.067 | 0.039 | 1.288 | 0.198 |
| Age | | 0.021 | 0.024 | 0.022 | 0.853 | 0.394 |
| Laws and norms favorable to drug use | Community | 0.053 | 0.068 | 0.021 | 0.785 | 0.433 |
| School opportunities for prosocial involvement | Community | 0.055 | 0.075 | 0.021 | 0.735 | 0.463 |
| Opportunities for prosocial involvement | Community | 0.009 | 0.040 | 0.006 | 0.228 | 0.820 |
| School–negative relations with peers | Social | -0.041 | 0.055 | -0.019 | -0.738 | 0.461 |
| Rewards for prosocial involvement | Community | -0.047 | 0.044 | -0.030 | -1.078 | 0.281 |
| Low neighborhood attachment | Social | -0.062 | 0.044 | -0.040 | -1.426 | 0.154 |
| Rewards for prosocial involvement | Family | -0.108 | 0.062 | -0.062 | -1.733 | 0.083 |
| Attachment | Family | -0.168 | 0.082 | -0.087 | -2.039 | 0.042 |
| (Constant) | | -2.616 | 0.602 | | -4.346 | 0.000 |

the interdependent nature of RF and PF and makes no assumptions about the functional form of the relationships. The statistical tool that serves such purpose is Multiple Correspondence Analysis. This analysis was contrasted with Multiple Regression.

Our results highlight the importance of the interdependence of multiple factors. Linear regression is very limited when accounting for such interdependence. In fact, the higher the interdependence among independent variables, the less reliable the regression coefficients, because of multicollinearity. In addition, linear regression imposes a linear functional form on the relationship between independent and dependent variables, which may not necessarily reflect reality. Finally, the way in which the regression line is estimated is sensitive to influential observations. While MCA cannot estimate the effect of one variable on another as regression does, it does not rely on assumptions of functional forms and fully incorporates interdependence. See [47]. In addition, outliers do not influence results because they become part of one of the categories.

An extensive discussion of the differences between MCA and linear regression is outside the scope of this paper, and we are not claiming that MCA is a better technique than Linear Regression. However, our results do suggest that extreme care should be taken when deriving prevention and policy implications from specific statistical techniques. We suggest that structural interdependence and qualitative techniques are highly informative because they do not rely on assumptions of functional forms and specific characteristics of the data (e.g., independence or normality) and that triangulation of methods is highly advisable. For instance, our findings regarding the perception of availability of drugs being a determinant factor of early initiation through both methods suggest that this is an RF that should be prioritized. MCA, however, does not provide information about moderating effects involving RF and PF. Multivariate regression techniques offer the possibility of constructing and analyzing interaction terms that capture such effects. Future work may attempt to combine the two approaches. Another potential limitation of using MCA is that the categorization of variables that are inherently continuous, such as age of initiation, requires cut-off points for making groups (i.e., categories) of participants. It is arguable that different ways of forming groups may yield different outcomes. If the data and theory allows it, sensitivity analyses using alternative categorizations may offer relevant robustness checks. In this work, we used cut off points that were both consistent with the literature and empirically practical. An additional consideration is necessary in relation to the cross-sectional nature of our data. Young adolescents that were classified as "never" along to "over 17" age of onset, may decide to use substances in the near future. Our data cannot capture such dynamic. An extension of this work, using a longitudinal design, may test the present results by tracing the behavior of a cohort of young adolescents (i.e., under 12) until they turn 17.

Regarding specific RF and PF, our analyses highlight that the effect of a perceived availability of drugs remarkably persists across different methodologies, suggesting that such perception must be prioritized in health policy. Note that such a perception is influenced by public policy at different levels including the way in which the police controls and takes action with respect to micro traffic and consumption. This result is consistent with the work of Tucker, Pollard, De La Haye, Kennedy and Green (2019) [48] who found that the perceived availability of drugs was related with higher rates of cannabis use among adolescents in the Check Republic, which displays the highest rate of substance availability in Europe, with an increasing prevalence of cannabis use per year. Thus, perception of availability must be taken into account in the debate on legalization and/or decriminalization of some substances, totally or partially. As it pertains to the other RFs and PFs, our results show a staggering contrast between Multiple Correspondence Analysis and Multiple Linear Regression. While MCA suggest that priority should be given to environmental-based prevention focused on community and social factors,

regression results suggest that the focus should instead be placed on family factors. The MCA results are consistent with the notion that early initiation, which occurs during early adolescence, is highly sensitive to broader contexts (e.g., social and community). In their decisions for the pursuit of identification vehicles outside the family, adolescents are experiencing a process of decreasing relevance of the family environment [49]. In Colombia, the scant research on RF and PF in connection to the age of onset is consistent to the results found by regression analysis. It is reported that the absence of close family members who use drugs and the existence of parental supervision is associated with greater resistance to initiate [50]. Furthermore, age of onset among Colombian adolescents was also found to be indirectly influenced by family conflict when serially mediated by negative emotions and sensation seeking [31]

Preventing SUDs is a major objective of health policy. In this paper, we focused on early initiation of drug use during adolescence, as a relevant behavioral antecedent of various types of SUDs. Delaying the age of onset of substance consumption may be one of the most effective ways to reduce SUDs. However, there are multiple causes and complex psychosocial dynamics behind the decision of a young teenager to start consuming drugs. This study contributes to understanding these causes by assessing early initiation relationship with multiple RF and PF in the adolescent's proximal and distal environment, using a statistical method that explicitly accounts for the interdependencies of all the factors studied. We provide new insights into the nature of the relationships between RF, PF, and early initiation as well as the application of a methodological tool to assess it.

## Supporting information

**S1 Table. Summary of RF and PF category cutting points.**
(XLSX)

**S2 Table. Multinomial ordered logistic regression of early initiation on RFs and PFs.**
(XLSX)

## Acknowledgments

We thank the municipalities of Cogua and Ubate near Bogota for their commitment and help with data collection in schools.

## Author Contributions

**Conceptualization:** Carlos Andres Trujillo, Diana Obando, Angela Trujillo.

**Formal analysis:** Carlos Andres Trujillo, Diana Obando, Angela Trujillo.

**Investigation:** Carlos Andres Trujillo, Diana Obando, Angela Trujillo.

**Methodology:** Carlos Andres Trujillo, Diana Obando, Angela Trujillo.

**Supervision:** Carlos Andres Trujillo, Angela Trujillo.

**Validation:** Carlos Andres Trujillo, Diana Obando.

**Writing – original draft:** Carlos Andres Trujillo, Diana Obando, Angela Trujillo.

**Writing – review & editing:** Carlos Andres Trujillo, Diana Obando, Angela Trujillo.

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
