## [Decision Letter · Decision Letter 0]

7 Aug 2019

PONE-D-19-16848

An examination of the association between early initiation of substance use and interrelated multilevel risk and protective factors among adolescents

PLOS ONE

Dear Dr. Trujillo,

Thank you for submitting your manuscript to PLOS ONE. After careful consideration, we feel that it has merit but does not fully meet PLOS ONE’s publication criteria as it currently stands. Therefore, we invite you to submit a revised version of the manuscript that addresses the points raised during the review process.

We would appreciate receiving your revised manuscript by Sep 21 2019 11:59PM. To enhance the reproducibility of your results, we recommend that if applicable you deposit your laboratory protocols in protocols.io, where a protocol can be assigned its own identifier (DOI) such that it can be cited independently in the future. For instructions see: http://journals.plos.org/plosone/s/submission-guidelines#loc-laboratory-protocols

We look forward to receiving your revised manuscript.

Kind regards,

Julie Maslowsky, PhD

Academic Editor

PLOS ONE

Journal Requirements:

Additional Editor Comments (if provided):

Thank you for your submission. The reviewers identified notable strengths of the work as well as a number of areas in need of improvement. Please respond in detail to each of the reviewers' comments.

Reviewers' comments:

Reviewer's Responses to Questions

**Comments to the Author**

1. Is the manuscript technically sound, and do the data support the conclusions?

Reviewer #1: Partly

Reviewer #2: Partly

2. Has the statistical analysis been performed appropriately and rigorously? 

Reviewer #1: Yes

Reviewer #2: Yes

3. Have the authors made all data underlying the findings in their manuscript fully available?

Reviewer #1: Yes

Reviewer #2: Yes

4. Is the manuscript presented in an intelligible fashion and written in standard English?

Reviewer #1: No

Reviewer #2: Yes

5. Review Comments to the Author

Reviewer #1: This study examined risk and protective factors associated with age of onset of substance use in a sample of adolescents from Colombia. Risk and protective factors were categorized within family, community, and social domains. The authors examined these factors specifically as predictors of early substance use onset using both Multiple Correspondence Analysis (MCA) and linear regression analysis. Results from this study suggest that community factors were most strongly related to early substance use initiation and that MCA may be an ideal analytic approach to studying multiple, interdependent variables. Although this manuscript has several strengths, including its data-driven approach to study the many potential factors influencing substance use initiation, the following concerns dampen my enthusiasm for publishing this manuscript in its present form:

Introduction

1. The authors indicate that risk factors will be abbreviated as “RF” and “FP”, but later use the full phrase (risk factors, protective factors) and “PF” throughout the manuscript. Please correct for consistency.

2. In the first paragraph, the sentence, “MCA is able to capture multiple associations among interdependent factors, without making assumptions about functional forms (linear or otherwise)” needs a citation to support this claim.

3. In the Introduction, and throughout the rest of the manuscript, sentence structure is at times awkward and confusing to read (e.g., the second sentence of the second paragraph starting with “Kessler et al….” and the fifth sentence of the second paragraph starting with “Chikritzhs…”). Having a native English speaker proofread the manuscript would be helpful in this regard. Also, please use decimal points rather than commas for the numbers you report in this study.

4. The section under “Risk and Protective factors associated with age of initiation” has several vague descriptions of prior findings that need additional specificity. For instance, the authors say “Kaplow et al. identified that children [childhood?] personal factors and parental patterns of drug consumption contribute to early initiation of drug use.” What are the “personal factors” in children that the authors are referring to? Personal factors could many mean different things (e.g., externalizing or internalizing behaviors, temperament, ADHD). Later in this section, the authors refer to “parental modeling” as a predictor of delayed alcohol initiation. It is unclear what behavior(s) the parents were modeling.

Methods

1. The description on how the survey was implemented should be moved to the beginning of the Methods section where research procedures are discussed.

2. What were the response options for each of the 17 risk and protective factors? In the supplement it appears as though the response scale ranges from 1-4, but it is unclear what each of these numbers equate to in relation to scale.

Results/Discussion

1. Was there any missing data across the variables examined in this study? If so, the authors need to report how they accounted for missing data in their analyses.

2. Since this study consisted of a cross-sectional sample of participants ages 12-19, mean age of 14, it is problematic to categorize substance use initiation as early, middle, and late. This is because the younger participants in this study may be misclassified. For example, a 12 year old in the sample who reported no use at the time of the assessment may begin use in the future by age 13 or 14 and therefore actually be an early initiator instead of a non-user. This limitation of the study needs to be addressed in the discussion section.

3. In the last section of the Results, the authors categorize certain factors as RF or PF. However, it seems like each RF could be a PF if the direction of effects were in the opposite direction. For example, “family conflict” is categorized as a RF. The authors should specify that high family conflict was a risk factor, because low family conflict could be a protective factor. Please add the direction of effects in this section.

Reviewer #2: The study propose a statistical novel approach to understand the interdependence of risk and protective factors of early initiation of drug use in Colombian Youth. Early initiation was define as the onset before the age of 15 years old (11-14 years). The authors posited that early initiation is an important predictor for drug disorders in later stages of life course. However, evidence is not conclusive about this causal relationship. Furthermore, the different drugs (i.e. alcohol, tobacco, cannabis) have different mean ages of onset. This could have implications for this kind of analysis. Overall, the analysis was well planned and provided the first evidence of this potential interdependence of factors from the Communities That Care Youth Survey and its adaptations worldwide. Also, contrast the MCA method with Multiple regression is considered by this reviewer a success and provide theoretical validation of the MCA findings for some factors such as accessibility to drugs. The importance of investigate the age of onset to prevent late disorders is clear, well structured and well supported by evidence.

Some risk and protective factors were mentioned with success. Their role as predictors helped to understand the potential causal chain between factors and short term outcomes.

In order to improve the manuscript and the analysis is recommended to consider the next minor issues:

1. Modify the title in order to mention that this analysis was performed in a Colombian youth sample

2. A section that explains potential interdependence between factors could be included

3.Also, no mention was made about the Colombian evidence in the topic ( if existed). The need of evidence in the field for Colombia should also be mentioned

4. Add references that indicate this potential factor interdependence

5. Is not clear if the used survey was just the Spanish version of the CTC youth survey or was the adapted version of the CTC-Youth survey for Colombia. If the Colombian version was used, the characteristics of the local adapted survey should be mentioned and referenced in order to provide the reader with information about the survey characteristics and its initial internal consistency –Cronbach's alpha Coefficient reported in 2015 (Mejía-Trujillo, Pérez-Gómez y Reyes-Rodríguez, 2015).

5. The multivariate regression model that will serve as comparison was not specified. The fitted equation should be explained in order to understand the model estimations that will be contrasted with the MCA

6. Descriptive statistics from table 1 are usually part of the methods section where the used instrument is described.

7. In table 1, risk and protective factors could be categorized by domain to provide better a understand of the table

8. The distribution of participants in the three early initiation categories should be shown. This could help to understand if the age ranges created do not affect the models. Moreover, it was mentioned that age 15 years is considered the cut point for EI. However, in Latinametican countries such as Colombia, the mean age of initiation is lower, therefore this age cut point should be validated. For instance the age of initiation associated with alcohol disorders in Colombia could be lower compared with the countries where <15 years was identified as the risky age. A an explanation of the potential different scenarios of using 14 or 13 as the cut point should be provided or discussed.

9. A better interpretation of Table 2 results could be: "The application of multiple correspondence analysis showed that the total inertia explained is equal to XXX (percent of inertia: 23.6% is due to the first axis dimension 1 and 13% due to the second dimension).

10. A remarkable simple explanation of the figures was made. In the figures the specific percentage of variance for each dimension should appear [e.g. Dim 1 (23.6%); Dim 2 (13.0%)]

11. It was not mentioned if interactions were examined. Significant interaction effects could modified the effects sizes. Also the fitted equation was not specified: Example: Y = βO + β1RFc + β2AGEc + β3GENDERc + β4RFc*AGEc+ β5RFc*GENDERc

Moreover, a multiple logistic regression could be considered for better comparison with MCA

12. There was no mention about how the authors dealt with missing data

13. "Regarding specific RF and PF, our analyses highlight that the effect of perceived availability of drugs remarkably persist across different methodologies suggesting that such perception must be prioritized in health policy."- This was a good conclusion. However should be confirmed after checking for interactions in the regression model

14. The manuscript did not include any limitations of the study. One limitation could be that despite this research contributes to understand these causes by assessing EI relationship with multiple RF and PF in the proximal and distal environment of the adolescent, no subgroup analysis could be done and was not discussed –due to the methods selected–.

6. PLOS authors have the option to publish the peer review history of their article (what does this mean?). If published, this will include your full peer review and any attached files.

Reviewer #1: No

Reviewer #2: No

---

## [Author Response · Author response to Decision Letter 0]

19 Sep 2019

Thank you for the opportunity to revise the manuscript. We are sending the revised version of the manuscript PONE-D-19-16848. We have addressed all comments and our answers are in the corresponding file "response to reviewers".

---

## [Decision Letter · Decision Letter 1]

29 Oct 2019

PONE-D-19-16848R1

An examination of the association between early initiation of substance use and interrelated multilevel risk and protective factors among adolescents

PLOS ONE

Dear Dr. Trujillo,

Thank you for submitting your manuscript to PLOS ONE. After careful consideration, we feel that it has merit but does not fully meet PLOS ONE’s publication criteria as it currently stands. Therefore, we invite you to submit a revised version of the manuscript that addresses the points raised during the review process.

We would appreciate receiving your revised manuscript by Dec 13 2019 11:59PM. To enhance the reproducibility of your results, we recommend that if applicable you deposit your laboratory protocols in protocols.io, where a protocol can be assigned its own identifier (DOI) such that it can be cited independently in the future. For instructions see: http://journals.plos.org/plosone/s/submission-guidelines#loc-laboratory-protocols

We look forward to receiving your revised manuscript.

Kind regards,

Julie Maslowsky, PhD

Academic Editor

PLOS ONE

Additional Editor Comments (if provided):

Please complete the following additional revisions:

1. The description of the analysis in the Methods section is not sufficiently detailed. Please move all text describing the steps of the analysis and the software used for analysis into the Methods section from its current position in Results.

2. Please add a note to each figure caption to aid in interpreting the figure without consulting the text. It is not immediately clear what 1-2, 2-1, 3-1, etc., indicate in the figure and the figures therefore do not stand alone.

3. The readability of the figures could also be improved through changing font size, size of the shapes on the plot, and using more distinctive shapes that will be distinguishable in black and white as well as color.

Reviewers' comments:

Reviewer's Responses to Questions

**Comments to the Author**

1. If the authors have adequately addressed your comments raised in a previous round of review and you feel that this manuscript is now acceptable for publication, you may indicate that here to bypass the “Comments to the Author” section, enter your conflict of interest statement in the “Confidential to Editor” section, and submit your "Accept" recommendation.

Reviewer #1: All comments have been addressed

Reviewer #2: All comments have been addressed

2. Is the manuscript technically sound, and do the data support the conclusions?

Reviewer #1: Yes

Reviewer #2: Yes

3. Has the statistical analysis been performed appropriately and rigorously? 

Reviewer #1: Yes

Reviewer #2: Yes

4. Have the authors made all data underlying the findings in their manuscript fully available?

Reviewer #1: Yes

Reviewer #2: Yes

5. Is the manuscript presented in an intelligible fashion and written in standard English?

Reviewer #1: Yes

Reviewer #2: Yes

6. Review Comments to the Author

Reviewer #1: (No Response)

Reviewer #2: Overall the manuscript represents a great effort to understand the importance of understanding the interdependence between RF and PF and their single effect early initiation of substance use. This version is clearest about how these potential relationships could occur and what are the potential implications on risk and protective factors prioritization. The authors addressed all previous comments successfully. In case those recommendations were not included, its rationale was satisfactorily discussed in the text. I recommend the manuscript publication.

7. PLOS authors have the option to publish the peer review history of their article (what does this mean?). If published, this will include your full peer review and any attached files.

Reviewer #1: No

Reviewer #2: No

---

## [Author Response · Author response to Decision Letter 1]

31 Oct 2019

Dear editor, we are resubmitting the mansucript addressing all your comments. Thank you

---

## [Editor Report · Decision Letter 2]

5 Nov 2019

An examination of the association between early initiation of substance use and interrelated multilevel risk and protective factors among adolescents

PONE-D-19-16848R2

Dear Dr. Trujillo,

We are pleased to inform you that your manuscript has been judged scientifically suitable for publication and will be formally accepted for publication once it complies with all outstanding technical requirements.

With kind regards,

Julie Maslowsky, PhD

Academic Editor

PLOS ONE
---

## [Editor Report · Acceptance letter]

12 Nov 2019

PONE-D-19-16848R2 

An examination of the association between early initiation of substance use and interrelated multilevel risk and protective factors among adolescents 

Dear Dr. Trujillo:

I am pleased to inform you that your manuscript has been deemed suitable for publication in PLOS ONE. Congratulations! Your manuscript is now with our production department. 

With kind regards,

on behalf of

Dr. Julie Maslowsky 

Academic Editor

PLOS ONE